# Effect of Power on Structural and Mechanical Properties of DC Magnetron Sputtered Cr Coatings

Wanyu Shi [1,2], Jian Peng [2], Zhigang Xu [3], Qiang Shen [2] and Chuanbin Wang [1,2,*]

1   Chaozhou Branch of Chemistry and Chemical Engineering Guangdong Laboratory, Chaozhou 521000, China
2   State Key Lab of Advanced Technology for Materials Synthesis and Processing, Wuhan University of Technology, Wuhan 430070, China
3   Hubei Key Laboratory of Advanced Technology for Automotive Components, Wuhan University of Technology, Wuhan 430070, China
*   Correspondence: chuanbinwang@whut.edu.cn

**Abstract:** Cr coatings were deposited on the surface of PCrNi1MoA steel by DC magnetron sputtering. The effects of power (100~250 W) on the structure and mechanical properties of the coating were systematically studied. The results show that all Cr coatings have strong (110) preferred orientation and anisotropic surface morphology, and the cross sections are columnar structures. With the increase in power, the deposition rate increases linearly, the surface roughness and grain size increase gradually, and the nanomechanical properties decrease first and then increase slightly. At 100 W, the columnar grain of the coating is compact and continuous, and the wear resistance and plastic deformation resistance are the best, with the highest binding force of 27.04 N. At 200 W, due to grain growth and tensile stress, internal defects increase, and mechanical properties are poor. The combination of coating and substrates is not tight enough, and it is easy to fall off and fail under the action of external force. When the power reaches 250 W, the higher bombardment rate helps the coating release stress, and reduce porosity.

**Keywords:** DC magnetron sputtering; Cr coatings; crystal structure; mechanical properties

## 1. Introduction

Metal Cr is a silvery-white metal with relatively stable chemical properties. The pure metal Cr coatings used for surface protection show stronger corrosion resistance, lower friction coefficient, and better high-temperature oxidation resistance [1]. The combination of electrocoating technology and metal Cr is the earliest, and it is widely used to prepare surface protective coatings in many countries because of its good economy and high technical maturity. As a result, chromium metal is listed as one of the most important coating metals, and in most cases, the chromium coating layer is specially used as the outermost coating of parts. Some barrels are coated with chromium only a few microns thick, but it remains after hundreds of rounds have been fired. Without the physical properties provided by chrome coating, the service life of most parts will be greatly shortened due to wear, corrosion, and other reasons and must be replaced or repaired frequently [2,3]. However, toxic gas released during electrocoating can seriously threaten health and environmental safety. At the same time, there are many inherent cracks in the electroplated hard chromium film, which affect the service performance [4,5]. Because of this, physical vapor deposition (PVD), which has a simple process and almost no pollution to the environment, is gradually coming into public view [6–9].

Magnetron sputtering technology is a kind of physical vapor deposition technology that originated in the early 1970s of the last century. The technology is developed on the basis of bipolar sputtering and has the advantages of high speed, low temperature, low damage, and low operating voltage, suitable for mass and high-efficiency industrial

production. Because the deposited coatings usually have excellent mechanical and physical properties, magnetron sputtering technology has been widely used in the fields of electronics, materials, optics, and semiconductors [10–12].

At present, many scholars have significantly improved the quality of the coatings by adjusting the sputtering process parameters [13–15]. Among them, the power greatly affected the structure and properties of Cr coatings. Li [16] studied the crystalline-amorphous transition of Cr coatings by DC magnetron sputtering. Preparation of Chromium coatings at different deposition rates by changing power. The results show that when the deposition rate decreases, the overall Cr coatings exhibit a mixed structure of amorphous phase and a small amount of nanocrystalline phase. When the deposition rate is maximum, the Cr coatings exhibit a dense columnar crystal structure. However, this study only explored the transformation of the structure of magnetron sputtering Cr coatings with power and failed to clarify the internal relationship between the structure and the properties of the coatings. Ferrec et al. [17] used HiPIMS to deposit Cr films that are isotropic and have no clear boundary. At all deposition pressures, increasing the peak power will reduce the surface roughness of HiPIMS-Cr films. Ferreira et al. [18] found that the peak power has a greater impact on the HiPIMS-Cr films. As the increase in peak power, there is always a (110) preferential orientation, but the deposition rate of HiPIMS-Cr films decreases. Higher power will reduce the grain size of the films and transform them from columnar to dense.

The above introduction shows that the change of power affects the structure and properties of the coatings. However, so far, the research on power is mostly focused on the HiPIMS-Cr coatings layer. To reveal the power impact on the structure and mechanical properties of DC magnetron sputtering Cr coatings, this study uses DC magnetron sputtering to deposit pure Cr coatings on the PCrNi1MoA alloy steel substrate, and then obtains the results through a series of mechanical properties tests for discussion.

## 2. Experimental

### 2.1. Coating Preparation

PCrNi1MoA alloy steel block with the size of 10 mm × 10 mm × 1 mm is selected for the substrates. The substrate, in turn, with different mesh sandpaper grinding, and polishing on the machine. Then, the substrate was ultrasonically shaken with acetone and ethanol solution for 15 min, respectively, and dried with a nitrogen blower. A high vacuum DC magnetron sputtering system (JCZK350A) was used for deposition. The chromium target (99.98% purity, $\varphi 50 \times 4$ mm) is mounted approximately 120 mm above the substrate. After the vacuum degree of the sputtering chamber was pumped to $9.9 \times 10^{-4}$ Pa by molecular pump, argon gas was injected into the chamber at a flow rate of 105.8 sccm until the Ar pressure in the chamber was maintained at 0.2 Pa. The substrate was heated to 500 °C by the resistance wire, and the deposition time was set to 120 min, and the sputtering power range was 100~250 W. Specific deposition parameters are shown in the Table 1 below.

**Table 1.** Deposition parameters for Cr coatings.

| Sampleno. | Ar Pressure (Pa) | Substrate Temp (°C) | Deposition Time (min) | Flow Rate (sccm) | Cathode Current (A) | Cathode Voltage (V) | Sputtering Power (W) | Thickness (μm) |
|---|---|---|---|---|---|---|---|---|
| Cr-1 | 0.2 | 500 | 120 | 105.8 | 0.417 | 240 | 100 | 2.77 |
| Cr-2 | 0.2 | 500 | 120 | 105.8 | 0.591 | 254 | 150 | 4.12 |
| Cr-3 | 0.2 | 500 | 120 | 105.8 | 0.760 | 263 | 200 | 6.04 |
| Cr-4 | 0.2 | 500 | 120 | 105.8 | 0.923 | 271 | 250 | 7.10 |

### 2.2. Characterization

The phase composition and orientation of the coatings were characterized using X-ray diffraction (XRD) applying an Empyrean (PANalytical, Malvern, UK, at 20 kV, 20 mA) equipped with Cu-K$\alpha$ radiation ($\lambda$ = 0.154 nm) in Bragg-Brentano configuration at a speed of 4 °/min and the scanning angle range is 40–90°. The microstructures of Cr coatings were observed with field emission scanning electron microscopy (SEM, JSM-6700F, Tokyo, Japan). In addition, the particle size analysis software Nano measurer was used to analyze the SEM images of the coating surface at the nanometer scale, in order to see the grain size and size distribution more directly.

NANOVEA ST400 (SENSOFAR, Shanghai, China) profiler was used to analyze the three-dimensional morphology and roughness of the coating surface, and the emission frequency was set at 1000 Hz. The selected area of each sample was 2 mm × 2 mm; the scanning step was 10 μm/s. The common evaluation parameters Sa and Sq were used as surface roughness to measure the roughness of the sample surface. Sa refers to the arithmetic average or geometric mean of the distance between the point inside the contour surface and the center plane, and Sq is the root-mean-square average of the ordinate coordinate of the roughness contour.

The hardness and elastic modulus of the Cr coatings on PCrNi1MoA alloys were investigated based on the Oliver and Pharr method with a Berkovich 142.3° diamond indenter. Each specimen was tested and averaged for ten indentations at 5 mN. The pressing depth was guaranteed to be less than 1/10 of the coating thickness to eliminate the influence of the matrix on the test results. The loading time, load holding time, and unloading time during the test were 5 s, 2 s, and 5 s. Using the measured hardness (H) and elasticity modulus (E), the ratio of H/E, and hardness (H) and the effective modulus of elasticity (E*, E* = E/(1 − $\upsilon^2$), $\upsilon$ = 0.3) is the ratio of $H^3/E^{*2}$ can be used to predict the toughness of the coating [19]. H/E is used to measure the elastic strain failure resistance of the coating, and $H^3/E^{*2}$ is used to measure the plastic strain deformation resistance of the coatings [20]. The higher the two values are, the stronger the elastic strain resistance/plastic strain resistance and the better the toughness of the surface coating.

The bonding strength of the coating was evaluated by qualitative and quantitative methods. The International VDI 3198 indentation test method was used for qualitative evaluation [21]. The diamond indenter of the Rockwell hardness tester is used to make a pit in the specified position of the sample. The test pressure load was 150 Kgf, and the load was maintained for 15 s. After unloading, the morphology around the indentation was observed under OM MX6RT (Sunny Optical Technology, Yuyao, China) optical microscope. The bonding strength was rated according to the radial crack and spalling, as shown in Figure 1. HF1–HF4 is qualified, and HF5–HF6 is unqualified. In order to quantitatively evaluate the bonding strength of chromium coating on PCrNi1MoA steel surface, diamond scratching experiments were carried out with WS-2005 (Anton-Paar, Lanzhou, China) coating automatic scratching tester. The diagram of the coating fracture is shown in Figure 2. The maximum load of the scratch test was 30 N, the loading rate was 25 N/min, and the stroke length was 5 mm. At the end of the test, the scratches are examined with a scanning electron microscope (SEM, JSM-6700F, Tokyo, Japan). In addition, according to the experimental results, the variation curves of AE and friction force with loading force are drawn, and more accurate binding force values are obtained.

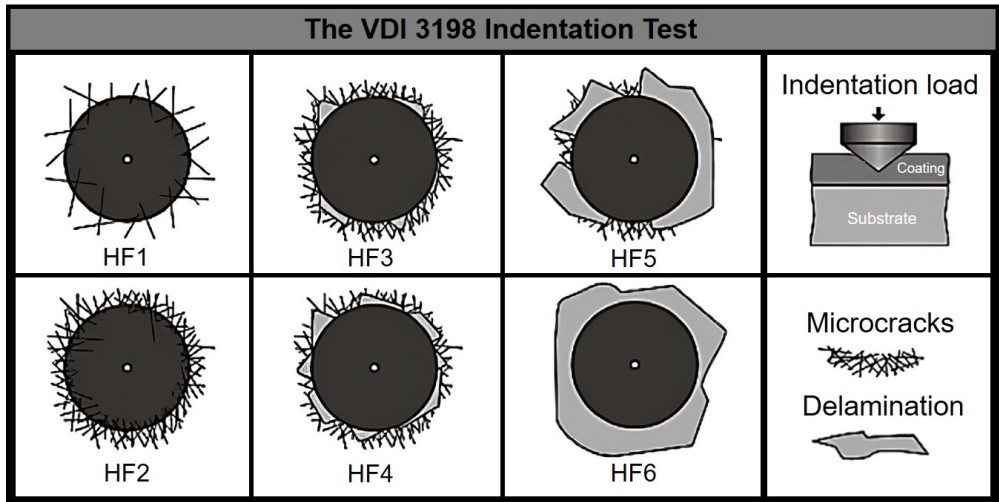

**Figure 1.** Combine strength rating criteria.

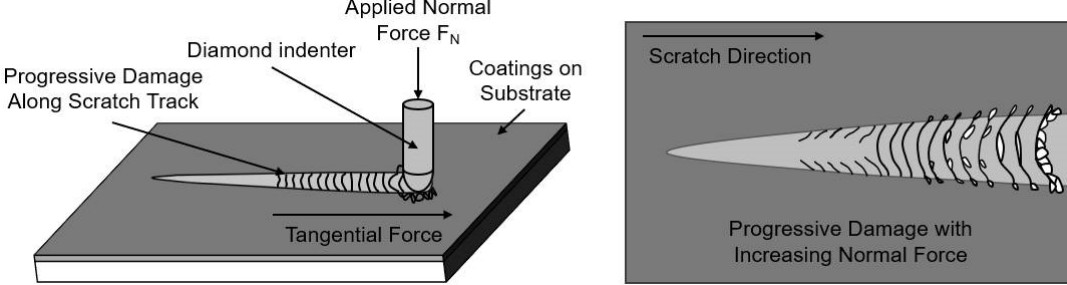

**Figure 2.** Diagram of coatings fracture.

### 3. Results and Discussion

*3.1. Deposition Rate*

The deposition rate of the coatings was calculated from the thickness divided by the deposition time (Table 1). The results show that the deposition rate increases linearly from 0.023 μm/min to 0.059 μm/min as the power increases from 100 W to 250 W. According to the report of Paturaud et al. [22], power is the most important process parameter affecting deposition rate. With the increase in power, the number of activated $Ar^+$ ions per unit of time increases, more target atoms are sputtered, and the average kinetic energy is higher. Thus, the deposition rate increases linearly with increasing power. Due to the high energy of sputtered particles, the deposited particles will migrate and fill the pores, densifying the coating. However, the density of the coating cannot increase indefinitely, and the excess energy is transferred to the substrate as heat dissipation. Thus, the density of the coating increases with increasing power until it approaches its theoretical value.

*3.2. Crystal Structure*

Figure 3 shows the XRD patterns of Cr coatings under different powers. The results show that the sputtered Cr coatings are all body-centered cubic structures with three main diffraction peaks of (110), (200) and (211). However, all Cr coatings show a preference for the (110) surface because the (110) has the lowest surface energy in the body-centered cubic structure. Cr coatings prepared by other magnetron sputtering technologies also have a similar crystal structure and preferred orientation [22].

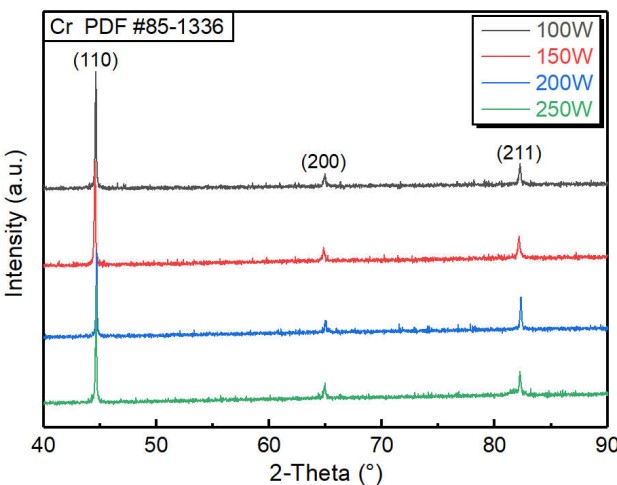

**Figure 3.** XRD patterns under different powers.

### 3.3. Microstructure

The SEM morphology and particle size distributions of Cr coatings at different powers are shown in Figure 4. The results show that the surfaces of all Cr coatings are composed of a triangular pyramid with different sizes, and the cross sections are composed of columnar crystal structures perpendicular to the surface of the substrate, with smaller columnar sizes near the interface. This is because the sputtering process completed the surface structure evolution of crystal nucleus formation, growing up and linking to form the coating [23,24]. Figure 5 shows the process of nucleation, growth, and thickening. Polycrystalline films are usually formed through the nucleation of isolated crystals on a substrate surface, and once formed, the nuclei lie laterally in the interface plane [25–27]. Lateral growth leads to the impingement and coalescence of crystals, resulting in the formation of grain boundaries. Grain coarsening can occur through the motion of grain boundaries resulting in the shrinkage and elimination of small grains, which, in turn, results in an increase in the average size of the remaining grains (Figure 5b). This is a well- known phenomenon called grain growth. With the strengthening of the bombardment effect, the number of ions sputtered increased, and the coating thickened gradually [28]. However, the grain boundaries formed by the crystal collision are fixed, and the grain structures generated by the nucleation, growth, and coalescence processes remain at the bottom of the coating. Usually, subsequent thickening occurs through epitaxial growth on these grains; as the grains at the base of the coating grow into the parent phase, competitive growth processes lead to an increasing in-plane grain size at the top surface of the thickening coating [29]. The larger grain size makes the sputtered atoms arrive at an angle that deviates from the normal of the coating surface, the shadow effect occurs, and voids form at the column wall. The competitive growth of grains is determined by their crystal structure. For magnetron sputtering Cr coating, grains can grow along three orientations of the body-centered cubic structure, but the surface energy required for (110) is minimal. As a result, most of the grains exhibit (110) preference orientation during coating thickening. In addition to minimization of strain and surface energy, anisotropy of growth rate also leads to grain competition on the thickened coating surface, as well as the evolution of in-plane grain size and through-thickness of average crystal texture, as shown in Figure 5c.

The growth structure of the coating depends heavily on the deposition rate during the whole growth process, and at a fast deposition rate, the structure shows a strong shadow effect (Figure 6). The power has the most direct effect on the deposition rate. Therefore, at low power (Figure 4a1), the columnar grains are compact and continuous, with no pores. With the increase in power, the columnar grain size increases, and internal defects are continuously produced. There are obvious grain boundaries between columnar structures, showing a porous columnar structure (Figure 4c1). However, when the power is 250 W, the atoms on the surface are more mobile due to the higher bombardment rate. The increase in

atomic mobility of the coating helps the coating release stress, so the crystallinity increases, the shadow effect weakens, the porosity decreases, and the columnar grains become dense and continuous again. On the other hand, at lower power, lower deposition rates contribute to the formation of coatings with less internal stress. The existing results show that the stress in chromium coating is generally anisotropic and has a great relationship with the coating thickness. The coating has less internal stress and a finer, denser structure, resulting in higher hardness.

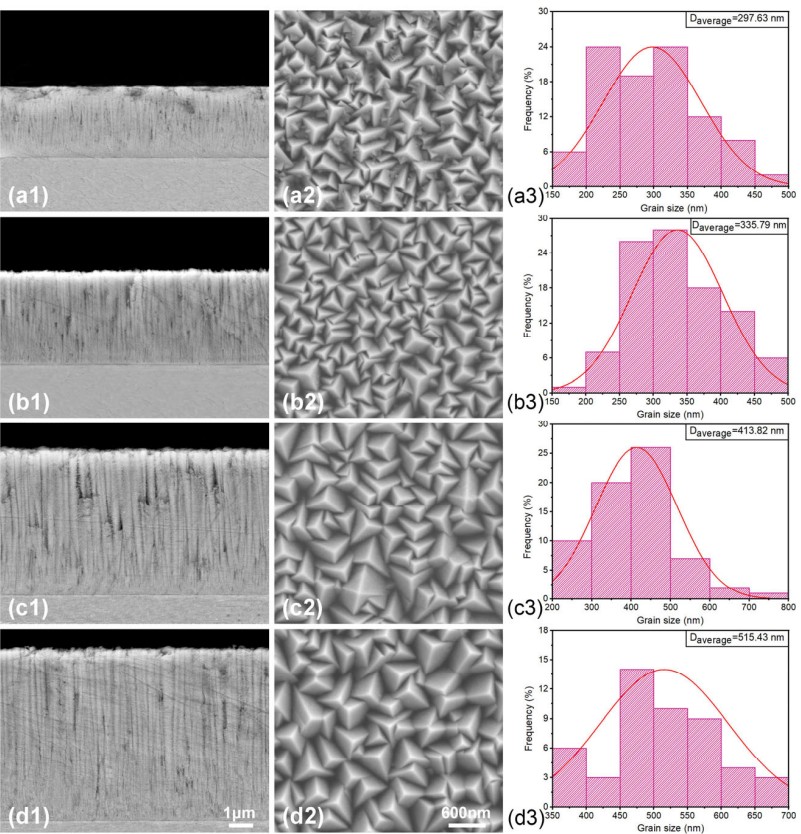

**Figure 4.** SEM micrographs and grain size diagrams of Cr coatings under different powers: (**a1**–**a3**) 100 W; (**b1**–**b3**) 150 W; (**c1**–**c3**) 200 W; (**d1**–**d3**) 250 W.

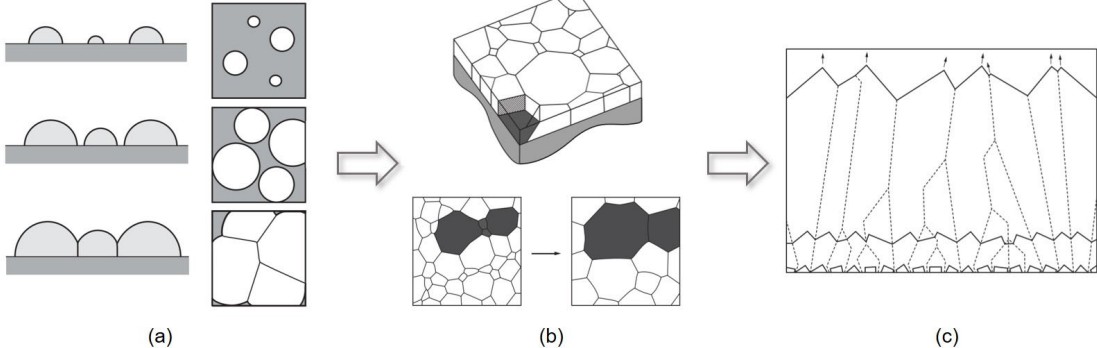

**Figure 5.** Overview of grain structure evolution during deposition of coatings: (**a**) nucleation; (**b**) growth; (**c**) thickening.

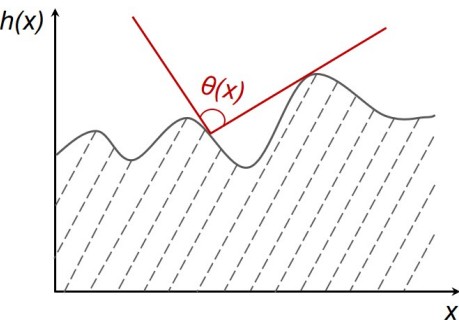

**Figure 6.** Shadow effect diagram.

### 3.4. Surface Roughness

Figure 7 shows the surface 3D profile and surface roughness of the chromium coating at different powers. The arithmetic mean value and root mean square value of surface roughness increase gradually with the increase in power. This is because when the power increases, the average grain size of the coating gradually increases from 297.63 nm to 515.43 nm. Due to the shadow effect, the triangular pyramidal crystal has pores, which makes the structure looser. The 3D surface morphology shows that the flatness of the coating surface decreases gradually with the increase in deposition rate, which is consistent with the conclusion obtained from the SEM micrographs. In conclusion, the surface roughness increases gradually under the influence of grain size increase and loose structure. When the power is 250 W, the maximum arithmetic mean value and root mean square value of the coating surface roughness are 0.015 µm and 0.027 µm, respectively.

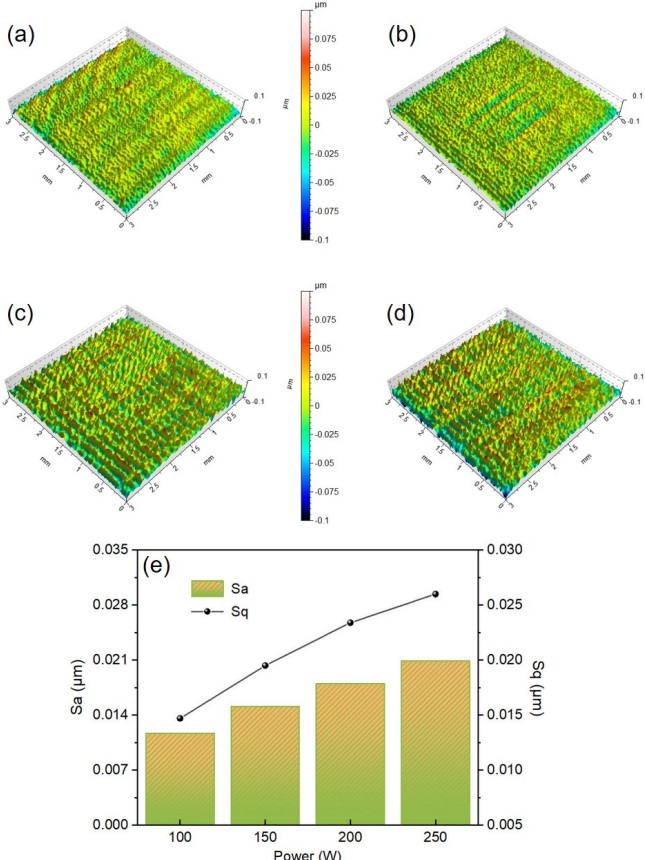

**Figure 7.** The 3D surface morphology of Cr coatings under different powers: (**a**) 100 W; (**b**) 150 W; (**c**) 200 W; (**d**) 250 W. (**e**) Surface roughness of Cr coatings under different powers.

*3.5. Nano-Indentation Behavior*

The nanomechanical properties of the coatings were tested by the nanoindentation instrument, as shown in Figure 8. Figure 8d shows that the pressing depth is within the range of 200−300 nm and does not exceed 1/10 of the total thickness of the coatings, which eliminates the influence of the substrate on the mechanical properties of the coatings using the following formula to calculate the elasticity of the coatings: $\eta = W_p/W_t$. Define the loading area under the curve (AOCB) as the total energy of the unloading process $W_p$, loading area to surround the close-unloading curve (AOB) as the total energy in the process of loading $W_t$; the calculation results are shown in Figure 8c. As the power increases to 200 W, the elastic recovery ratio also increases gradually, from 0.68 to 0.76. The interior of the coatings obtained by physical vapor deposition is usually in a state of stress, which is the force appearing in the interior of the sedimentary layer without external force and temperature field and is called internal stress. Internal stresses are divided into tensile and compressive stresses, the former usually positive and the latter usually negative. Fernandes et al. [30] showed that tensile stress would reduce the hardness of coatings, while compressive stress had the opposite effect. Lintymer et al. [31] showed that the porosity of the coatings also had a certain influence on the mechanical properties. As shown in Figure 8a, with the increase in power, the elastic modulus and nanoindentation hardness first decrease and then slightly increase. In addition, the ratio of hardness and elastic modulus (H/E) can be used to predict the elastic deformation resistance of the coatings, and $H^3/E^2$ can be used to measure the plastic strain resistance of the coatings [32]. The higher the two ratios are, the stronger the resistance to elastic/plastic strain and the better the toughness of the surface coatings [33,34]. Figure 8b shows the H/E and $H^3/E^2$ ratios of the coatings. The variation trend of this value is the same as that of H and E, which is consistent with the research results of the elastic recovery ratio. The H and E values of the coatings prepared at 100 W are the highest, and the deformation resistance is the strongest. According to the SEM image, the grain size of the coatings under this condition is the smallest, and the columnar grains are compact and continuous. In general, smaller grain sizes increase grain boundary fractions and provide an additional barrier to lattice dislocations, resulting in increased strength, as well as a tight bond between columnar grains and a low number of defects that reduce the likelihood of cracking through the thickness. In addition, the induced compressive stress can make the coatings withstand the greater tensile strain, which helps to enhance the toughness. In summary, the increase in hardness and toughness is a synergistic effect of dense microstructure, refined grain size, and an appropriate level of compressive stress increase. The hardness and toughness of the coatings prepared at 200 W were the worst. Because with the increase in high-energy bombardment ions, the compressive stress at the grain boundary is gradually offset by the tensile stress generated by point defects, and the hardness decreases. In addition, the high porosity of the columnar grain of the coatings also leads to a low elastic modulus. However, when the power reaches 250 W, the hardness and toughness of the coatings increase slightly. With the further increase in the bombardment level, the chromium coatings morphology becomes denser. Although there are still some columnar remnants, the pore level is reduced, the hardness increases to about 3.45 GPa, the elastic modulus increases to 108 GPa, and the elastic recovery ratio also increases.

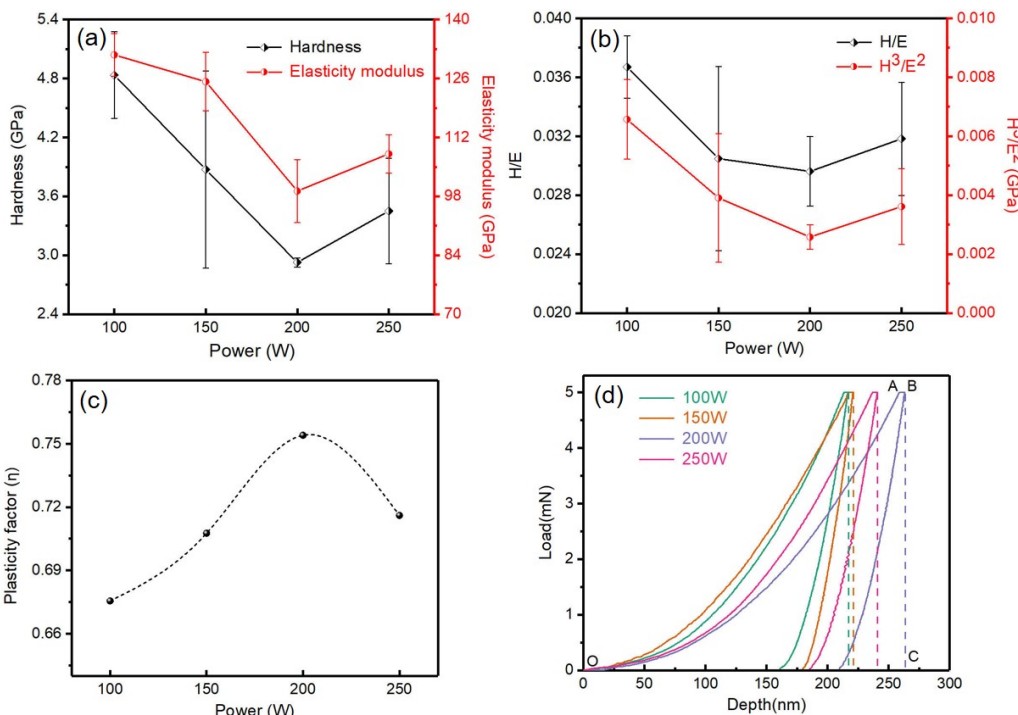

**Figure 8.** (**a**) Hardness and elasticity modulus, (**b**) H/E and H$^3$/E$^2$, (**c**) plasticity factor, and (**d**) load-depth curves of the Cr coatings under different powers.

### 3.6. Micro-Scratch Behavior

Figure 9 shows the Rockwell indentation optical microscope photos of Cr coatings at different powers. Due to the influence of power on deposition rate, the coating thickness varies greatly. When the coating is thick, the scratch test cannot penetrate the coating to the substrate under the maximum load, so the effective binding force data cannot be obtained. The indentation test is an international method to evaluate the adhesion of coatings with different thicknesses. As can be seen from the figure, only when the power is 200 W, does the coating breaks and falls off in a large range, and the binding strength is only HF4 level, indicating that the coating has basically failed at this moment. Under the remaining power, there is no shedding around the indentation of the coating, and only a few cracks are observed. Therefore, the bonding strength reaches the HF1 level, indicating that the coating and substrates are closely bonded. It is worth noting that when the power is 100 W, there is almost no crack near the indentation. In order to obtain the film base binding force of the coating with the best performance more directly, the quantitative evaluation of the coating was carried out by micro-scratch test. Figure 10 shows the change of friction force and sound signal with the increase in loading force when the power is 100 W. $L_{c2}$ is the critical point of cohesive failure of the coating; that is, an intergranular fracture occurs inside the coating under the action of load. However, the coating will not come off, and the substrate will not be exposed. The value of $L_{c2}$ is 27.04 N, indicating that when the power is 100 W, the coating is closely bound to the substrate and can play a better protection role for the substrate under the action of external force. This may be related to the smaller size of the columnar crystal, the lower pore level, and the stress distribution inside. With the increase in power, the columnar grain porosity of the coating is too high, the toughness is poor, the fracture is easier to occur, and the bonding strength is poor. The fracture properties of Cr coatings at different powers are consistent with the above hardness and toughness test results.

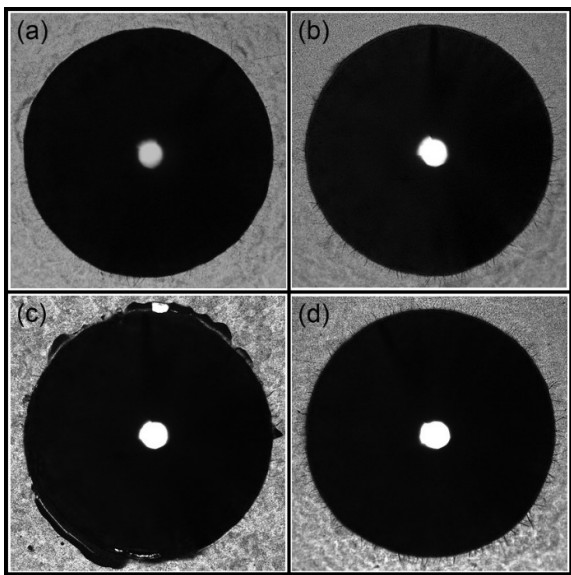

**Figure 9.** Rockwell indentation of Cr coatings at different powers: (**a**) 100 W; (**b**) 150 W; (**c**) 200 W; (**d**) 250 W.

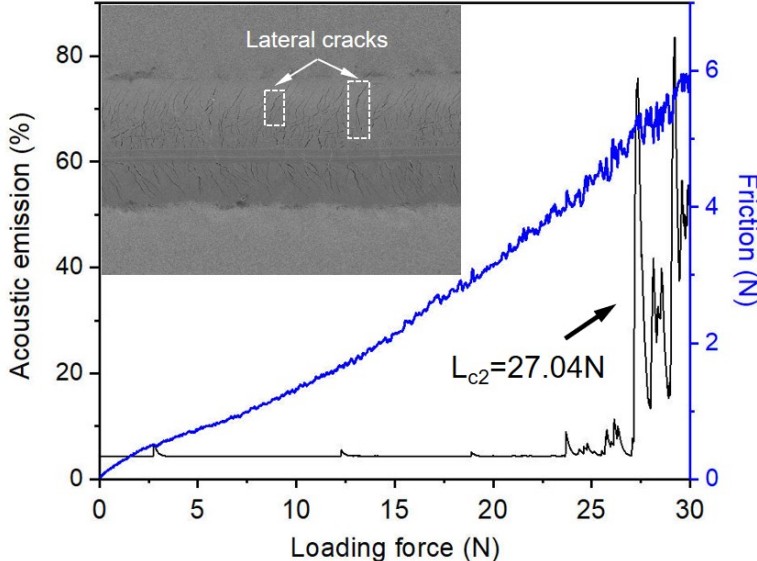

**Figure 10.** Scratch morphology and acoustic emission curve of coating at 100 W power.

## 4. Conclusions

Chromium coatings were prepared by DC magnetron sputtering with different power on the surface of PCrNi1MoA steel. The deposition rate increases linearly from 0.023 µm/min to 0.059 µm/min as the power increases from 100 W to 250 W. Three major diffraction peaks (110), (200), and (211) of Cr can be identified in the XRD figure. All Cr coatings have a (110) plane preferred orientation crystal structure. The power has no obvious effect on crystal orientation. The chrome coating grows into a columnar structure with a three-sided tapered tip. At low power, the columnar grains are compact and continuous, with no pores, good mechanical properties, best wear resistance and plastic deformation resistance, and close binding with the substrates. With the increase in power, the surface roughness and grain size increase gradually, and the internal defects appear constantly. The tensile stress caused by higher porosity leads to the decrease in hardness and elastic modulus of the Cr coating, which is more prone to intergranular fracture and falls off under the action of external forces. When the sputtering power reaches a certain level, the higher bombardment rate

can help the coating release stress, reduce porosity, restore the mechanical properties of the coating, and make the intergranular bonding closer.

**Author Contributions:** Conceptualization, W.S.; methodology, W.S.; software, W.S.; validation, W.S., J.P. and Z.X.; formal analysis, Z.X.; investigation, C.W.; resources, Q.S.; data curation, C.W.; writing—original draft preparation, W.S.; writing—review and editing, W.S. and C.W.; visualization, J.P. and Z.X.; supervision, Q.S.; project administration, C.W.; funding acquisition, C.W. All authors have read and agreed to the published version of the manuscript.

**Funding:** This research was funded by the Guangdong Major Project of Basic and Applied Basic Research (2021B0301030001), the National Key R&D Program of China (2021YFB3802300), and the Self-innovation Research Funding Project of Hanjiang Laboratory (HJL202012A001, HJL202012A002, HJL202012A003). And the APC was funded by the Guangdong Major Project of Basic and Applied Basic Research (2021B0301030001).

**Institutional Review Board Statement:** Not applicable.

**Informed Consent Statement:** Not applicable.

**Data Availability Statement:** The authors confirm that the data supporting the findings of this study are available within the article.

**Conflicts of Interest:** The authors declare no conflict of interest.

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
