# Peer review of "Effect of Power on Structural and Mechanical Properties of DC Magnetron Sputtered Cr Coatings"

_metals, doi:10.3390/met13040691_

Round 1

Reviewer 1 Report

A few comments to the article.

1) Figure 3. can be deleted. It contains one straight line. It is enough to describe this dependence in one sentence: With the increase of power from 100 to 250 W, the deposition rate increases linearly from 0.025 to 0.06 um/min.

2) The current and voltage of the magnetron discharge are not specified. Only power. This deprives readers of useful information, because for different experimental conditions (pressure, target thickness) these parameters can change at constant power.

3) Target thickness is not specified.

4) " The chromium target (99.98% purity) is mounted approximately 120 mm above the base bracket. "

What does this mean? Target-to-substrate distance?

5) The dependencies shown in Figure 7 have an ambiguous character of non-monotonic dependence, which does not allow us to draw certain conclusions due to the small number of experimental points. For example, in the range of 100-200 W, plasticity increased. But at the power of 250 W, it decreased. What's it? Reverse trend or statistical error? To answer this question, there is definitely not enough experimental data at the following points, for example, at the 300 Watt.

6) The correct position of the captions to Figures 7 and 8 is required.

Reviewer 2 Report

This paper is written very confuse with a lot  of sentences which are unclear. For the beginning, English must be substantially improved. Authors must make additional effort to improve their manuscript giving deeper scientific significance of the presented results. This version looks like as technical report, not as original scientific article. There is also confusion with images. The aim of the paper must be better underlined.

See specific comments on the paper:

1) Abstract:  The sentence: "When the power reaches a certain level, the higher bombardment rate helps the coating release stress, reduce porosity". The certain level must be indicated.

2) Introduction: What Authors want to say with "High strength steel also has high strength .... ".

The corresponding references inside the first two paragraphs must be given.

3) Experimental: What is HRC? What is HF?

Results and discussion:

Section 3.1. The thickness of coatings obtained with a different of power should be given. Authors should supply cross section analysis of their systems to give the coating thickness.

Section 3.2. The corresponding JCPDS standard for Cr should be given to show the development of (110) preferential orientation in the coatings.

Section 3.3. Which is a difference between Sa and Sq roughness? Which is physical meanings of these roughness?

Authors must correlate morphology of their coatings with crystal structure to give deeper scientific meaning to their investigation.

4) The reference list is inappropriate and must be extended.

Reviewer 3 Report

The aim of the paper is to assess the effect of power (100 ~ 250 W) on the structure and mechanical properties of Cr coatings deposited by DC sputtering. Although the topic is interesting and compatible to the journal scope, the paper lack in some aspects and it need to be revised, rearranged and improved. Some considerations are in the following detailed.

-        The introduction need to be revised. Several researches in the literature are specifically focused on power effect on DC sputtering of Cr based coatings. A detailed description of the research improvement in this direction is mandatory. Furthermore, the novelty of the proposed paper compared to the literature references need to be clearly addressed. This is the critical point of the paper, also corroborated by a not clear evidence in the result and discussion section of the scientific soundness of the paper. All the results and discussion section need to be revised to better calry the novelty output of the paper.

-        Figure 3. Replicas and standard deviation need to be added in the plot and discussed in the manuscript.

-        Figure 5. The optical profilomenter is not the ideal option to assess the correlation between the roughness and crystallography. AFM is the ideal option. Please on the measurements add also info on error of data.

-        Figure 6. Discuss about coating thickness. Please add results of coatings in thickness. Apparently a transition from small to large thickness takes place at 150-200W. The crystal size distribution is not discussed in the text. The average crystal size not grows with increasing power. A3 and B3 plots show a bimodal trend. All these aspects need to be discussed.

-        Figure7. The Authors related microhardness measurements with microstructure and crystal size. Comparing Figure 6 and 7 some issues are evident. Considering the non linear trend of average crystal size. This aspect need to be better discussed. If required, new SEM measurements able to validate the obtained crystal size distribution could support the Authors to define affordable hypothesis (e.g. error bar at 150W is high).

-        Figure 8. A mistake takes place on location of Figure7 images and figure 8 (image and caption). Rearrange them. Without all photo it is impossible to add comments in this section.

-        Please, justify why the “Rockwell indentation of Cr coatings” discussion is in Micro-scratch behavior. If I understood Micro-scratch behavior was possible to perform only on 100W sample, avoiding the possibility to  assess a trend. All this section need to be improved. It should/could be the novelty point of the paper and at this stage not give relevant improvement of knowledge on the paper. 

Round 2

Reviewer 2 Report

Authors adopted all my remarks, and this improved version of the manuscript can be accepted for publishing.

Author Response

Thank you so much for your great patience and huge help. Best wishes to you.

Reviewer 3 Report

before reviewing the article, the latest version of the manuscript must give evidence of the revisions applied by the authors. Without this latter it is not possible to apply an effective review of the article. From a my revision of the manuscript, the improvement of the paper is very limited. The comments were not managed or deeply addressed (some points were referred to data incongruences).  

Round 3

Reviewer 3 Report

In Figure 4 I think that we have a mistake with b3 and c3 changed.  Th Davg indicated in b3 is referred to c3 hystogram and vice versa. Why do we have a bimodal trend in two hystograms?. In fact, the Gaussian not fit two hystograms (effects due to crystals germination and growth?). This point need to be discussed in the article.  
